



# Do low-cost ceramic water filters improve water security in rural South Africa?

Jens Lange[1], Tineke Materne[1], Jörg Grüner[2]

[1]Hydrology, Faculty of Environment and Natural Resources, University of Freiburg, Freiburg, 79098, Germany

[2]Forest Botany, Faculty of Environment and Natural Resources, University of Freiburg, Freiburg, 79098, Germany

*Correspondence to*: Jens Lange (jens.lange@hydrology.uni-freiburg.de)

**Abstract.** This study examines the performance of a low-cost ceramic candle filter system (CCFS) for point of use (POU) drinking water treatment in the village of Hobeni, Eastern Cape Province, South Africa. A stepwise laboratory test documented the negative effects of repeated loading and ambient field temperatures. Moreover, CCFS were distributed in

Hobeni and a survey was carried out among their users. The performance of 51 CCFS was evaluated by dip slides and related to human factors. Already after two thirds of their specified lifetime, none of the distributed CCFS produced water without distinct contamination and more than one third even deteriorated hygienic water quality. Besides the water source (springs were preferable compared to river or rain water), a high water throughput was the dominant reason for poor CCFS performance. These findings suggest that not every CCFS type per se guarantees improved drinking water security and that

the efficiency of low-cost systems should continuously be monitored. For this purpose, dip slides were found to be a cost-efficient alternative to standard laboratory tests. They consistently underestimated microbial counts but can be used by laypersons and hence by the users themselves to assess critical contamination of their filter systems.

**Keywords**

Point of use, drinking water treatment, ceramic candle water filter, dip slides, developing region, coliform bacteria, South Africa

## 1 Introduction

Within the last 25 years, 2.6 billion people have gained access to improved drinking water, while 663 million are still

threatened by unsafe water from surface sources, unprotected springs or wells (WHO, 2015). Nearly half of the people without access to improved drinking water live in Sub-Saharan Africa and most of them in rural areas. In South Africa, for example, only 78% of the rural population had access to improved drinking water sources compared to 99% in urban areas, and over 3000 deaths due to unsafe water were estimated in 2012 (WHO, 2014).



In rural areas, decentralized water treatment systems provide opportunities to improve water security. Their efficiency against bacterial, viral and protozoan pathogens has widely been documented (e.g. Sobsey et al., 2008; Peter-Varbanets et al., 2009). Methods include application of heat or UV (boiling, solar radiation, UV lamps), chemical disinfection (e.g. by chlorine or silver), and physical removal by reverse osmosis or filtration (using activated carbon, granular media,
membranes, ceramic cups or fibers). Advanced physical methods (like reverse osmosis or ultrafiltration) are safest and most efficient but largely limited to industrialized countries due to high investment costs. First projects in developing nations installed them as small scale systems supplying purified drinking water to entire villages or small communities. Modern stations can run on solar power and contain storage tanks for periods of unfavorable sunlight conditions (Sima & Elimelech, 2013; Elasaad et al., 2015). However, the treated water must still be transported and stored in households after treatment,
which leads to a risk of recontamination (Gundry et al., 2006). Hence, point of use (POU) household water treatment may contribute to water security also in these cases. Among various methods, ceramic and biosand household water filters were identified to be most effective (Brown, 2007; Sobsey et al., 2008). For rural South Africa, low-cost ceramic water filters have been advocated for POU water treatment (Du Preez et al., 2008; Mwabi et al., 2013), although a recent study questioned their long term efficiency under field conditions (Mellor et al., 2014). Then human factors are crucial for POU
water treatment, since regular maintenance and adequate cleaning are preconditions for microbial efficiency. As a consequence, a specific filter type that proved efficient in the laboratory, might totally fail when distributed to rural communities and actually used under field conditions. This is particularly true for low-cost systems.

To prevent the failure of well-intentioned development projects, the efficiency of water filters should therefore continuously be monitored. Dip slides are a suitable way to comply with the need to frequently monitor aseptic conditions or treatment
success of disinfection systems. Examples include the hygienic control of endoscopes in hospitals (Gerstenberger, 2008) or of cooling lubricants during processing of metals in mechanical engineering (Barth, 2003). Joyce et al. (1996) used dip slides under field conditions in Kenya to study the efficiency of water disinfection by solar heating but did not document their reliability. Later, Sandhya et al. (1999) introduced dip slides as a robust, quick and cost-effective method to qualitatively evaluate E. coli contamination down to concentrations of $10^2$ coliform units (CFU) $l^{-1}$ in water. Though limited in accuracy,
dip slides are much cheaper than standard laboratory tests and can also be used by unskilled personal.

This study combines laboratory and field investigations to examine the efficiency of a widely used low-cost Ceramic Candle Filter System (CCFS) for POU water treatment. The microbiological efficiency of the CCFS is examined by a stepwise laboratory test. Moreover, CCFS systems are distributed within a remote, rural area in South Africa and a survey is carried out among their users. Thereby, CCFS performance is evaluated by dip slides after an eight months use and related to human
factors.



## 2 Study area

The study area is located in the village of Hobeni, Mbhashe municipality which is situated in the south eastern part of the Eastern Cape Province, South Africa (Fig. 1). Climate is temperate with 18.3°C mean temperature and annual rainfall of 1041 mm. Mbhashe municipality is considered a remote, rural area with difficult conditions regarding water supply (Momba

et al., 2006). About 50% of its inhabitants use untreated surface water from unprotected rivers or springs as their main drinking water source, 16% harvest rain water. During the study period, only one household of Hobeni village had access to treated water from communal water taps, none had its own groundwater well.

## 3 Methodology

### 3.1 The Ceramic Candle Filter System (CCFS)

The investigated two-bucket CCFS is commercially distributed under brand the name DrinC by Headstream Pure Water Johannesburg, South Africa. It consists of the candle filter unit, wedged between two 20 l plastic buckets (Fig. 2). A tap is inserted at the base of the bottom bucket, which technically represents a safe storage container for drinking-water according to local standards (CAWST, 2011). The candle filter unit consists of a silver-impregnated ceramic shell containing an activated charcoal interior medium. The main parts of the CCFS can be produced locally, only the filtering candles are

imported from Europe or Northern America. With flow rates of approximately $1\,l\,h^{-1}$, depending on the batch volume, CCFS are able to produce an adequate daily drinking-water volume (CAWST, 2011). Filtering candles have to be replaced after 12 months (DrinC, 2016).

### 3.2 Dip slides

Nutrient TTC-MacConkey (NUT/MAC) dip slides (Precision Laboratories, 2016) were used to study the performance of the

CCFS under field conditions. They consist of a plastic paddle with different agar media on each side, covered by a plastic vial. Nutrient-TTC Agar supports the growth of a wide range of bacteria, while MacConkey Agar identifies lactose-fermenting coliforms. For testing, the paddle is removed from the vial, dipped into the water for 15 to 20 s, incubated for 24 h at 36 ± 4 °C, and evaluated against colour charts provided by the manufacturer. Minimum precision is $10^4$ CFU $l^{-1}$. An application software for mobile devices (BioPaddlesLite©) has recently been published to evaluate dip slide readings with

the help of standard images.

### 3.3 Stepwise laboratory performance tests

Laboratory performance of the CCFS was evaluated in the hydrology laboratory of the University of Freiburg, which is accredited for microbiological drinking water analysis according to German DIN EN ISO standards. In these standards, E. coli bacteria are regarded as efficient indicators for faecal pollution (Paruch & Mæhlum, 2012). For CCFS performance tests



they were used in reference solutions at concentrations of approximately from $10^{13}$ to $10^{16}$ CFU l$^{-1}$. The top buckets of four CCFS were filled with 20 l of tap water and spiked with 5 ml of reference solution. Background contamination was excluded by blank samples taken from the taps and the buckets. The initial bacterial concentration in the top buckets was evaluated by 100 ml samples taken by a sterile, graduated pipette. Testing started at ambient temperatures of 21 °C. The first laboratory

protocol included a low filling scenario: the CCFS were filled once and 100 ml of filtrate was sampled after 7, 24 and 48 h. This was followed by a second laboratory protocol, which included a high filling scenario: during three days the top bucket was filled every day and 100 ml filtrate was sampled 7 h after each filling. Afterwards, the CCFS were placed inside a laboratory incubator at 27 °C to simulate field conditions and both a high and low filling scenario was conducted following the above protocols. Between all filling scenarios the CCFS buckets were rinsed with tap water and cleaned with a medical

surface disinfectant. The collected samples were analysed by the membrane filter method according to German DIN EN ISO 9308 drinking water standards. In parallel, NUT/MAC dip slides were used.

### 3.4 CCFS distribution and field survey

In June 2014 a community meeting was hold in Hobeni, where the CCFS were described in detail and distributed among 150 rural households that showed interest. In January/February 2015, 91 households were visited during a three week period. The

15 demographics of each household (number of adults/children living in the household, education of household members), the general acceptance of the CCFS (positive, negative) and aspects of maintenance and use (filling frequency, cleaning, water sources) were evaluated by a questionnaire. In 51 households CCFS were still in use. Their performance was evaluated comparing bacterial contamination of the top and bottom buckets by the NUT/MAC dip slides. In the top CCFS buckets the paddle was removed from the vial and dipped into the water for a contact time of 15 to 20 s with gentle stirring. The bottom

buckets were tested by directly filling filtrated water from the CCFS tap into the dip slides vials that were subsequently tested by the paddles for 15 to 20 s. After the tests, vials were emptied, paddles and vials of all dip slides were closely linked, placed in a thermos bag and incubated after no more than 7 h at 36 ± 4 °C using a mobile incubator. Ambient air temperature, as well as temperature and electrical conductivity inside the CCFS buckets were measured by a portable conductivity meter PCE-PHD 1 (PCE Instruments).

## 4 Results

### 4.1 Laboratory performance tests

The first low filling scenario at 21 °C ambient temperature resulted in 100% removal of the bacteria as detected by the membrane filter method (Fig. 3). Not a single coliform unit could be detected in the filtrate after 7, 24 and 48 hours. Also the high filling scenario suggested satisfactorily CCFS performance, although single coliform units were detected in the filtrate.

However, during later tests at ambient temperature, typical for field conditions, no filtrate was free of coliform units and maximum values reached $5 * 10^3$ CFU l$^{-1}$. Although removal rates were still above 99.99%, these findings suggest that both



the repeated loading and the higher ambient temperature affected CCFS performance. As a consequence, the CCFS did not produce water that complied with international standards for drinking water (WHO, 2011). We did not find significant differences between the filling scenarios or immediate effects of different loading concentrations. Both the membrane method and the NUT/MAC dip slides attested microbial purity of the drinking water. Due to their limited precision, the dip
slides detected contamination only in one out of four bottom buckets during the final filling scenario (Fig. 4). In the top buckets, the accredited membrane method yielded coliform counts ($2.8 * 10^9$ - $1.7 * 10^{11}$ CFU $l^{-1}$) that were underestimated by the dip slides ($3.8 – 6.0 * 10^6$ CFU $l^{-1}$). Although there was no clear linear relationship between the two methods, we did not obtain false positive dip slide readings.

**4.2 Field survey**

Hobeni is a small rural community with low average income. None of the 91 households that were visited had access to piped water at the own dwelling and only 26 % to toilet facilities. On average, households consisted of 6 people and used water from different sources to fill the CCFS (Fig. 5). The acceptance of the CCFS was generally high. Approximately eight months after distribution, 69 % of the CCFS were still in regular use and 60 % of the households liked the clean water after the filtering procedure. The majority (84%) declared no incidences of digestive afflictions, including diarrhea, in the family
during the last five years. This was supported by an official statement of the Hobeni Clinic. However, the microbiological water quality of 51 tested CCFS showed contrasting results: none of the filtrate samples was free of coliform bacteria and 35 % even showed a deterioration of water quality (Fig. 6). To evaluate the reasons for the CCFS-failure in Hobeni, the CCFS-efficiency was evaluated as percentage of remaining coliform units in the filtrate and plotted against various household characteristics evaluated in the questionnaire survey. We excluded one outlier that showed a massive deterioration (>1400%;
Fig. 6). All regressions showed a large scatter and $r^2$-values were below 0.1, which precluded quantitative statistical analysis (Fig. 7). Nonetheless, the direction of the influence could visually be determined. Similar to the laboratory performance test, the ambient temperature showed an influence on CCFS efficiency. This was more prominent for water temperature inside the system than for ambient air. Only when the water temperature exceeded 21 °C, some of the CCFS deteriorated water quality. A second factor was the intensity of CCFS use. This was evident from the filling frequency and from the numbers of people
living in the households. Here the number of children had stronger effects on CCFS efficiency than the number of adults. More frequent cleaning could obviously not compensate for frequent CCFS use. A third factor was the water source: spring water apparently had higher quality. This could be traced by electrical conductivity, since values measured in CCFS only containing spring water ($0.27 \pm 0.13$ mS $cm^{-1}$) were approximately double than in those that were also filled by rain and river water ($0.13 \pm 0.14$ mS $cm^{-1}$). While in-door keeping of animals showed a weak influence, our data did not suggest a positive
effect of the educational level of CCFS users.



## 5 Discussion

The dip slide testing method was chosen for field testing because of the lack of access to a laboratory in the remote study area. The commercial product we used was only sensitive to concentrations down to $10^4$ CFU $l^{-1}$ and therefore not recommended for drinking water. On the one hand, this was corroborated by our laboratory test, where low CFU counts

could not be detected. A consistent underestimation of dip slides is known (e.g. Kinneberg & Lindberg, 2002), which limits the quantitative evaluation of dip slide readings and also produced a large scatter in our data (Fig. 7). On the other hand, we did not obtain any false positive dip slide reading in our laboratory test. This makes NUT/MAC dip slides a valid compromise to assess the efficiency of drinking water treatment in our remote study area. We consider them appropriate to determine critical microbial presence in drinking water at minimal cost. They can directly be used in the households and in

principle also directly by the CCFS users. New developments in free evaluation software for mobile devices will even increase their acceptance and reproducibility in future. Since only a regular and repetitive monitoring guarantees the success of POU water treatment, we recommend that treatment systems should be accompanied with cost-efficient monitoring techniques where dip slides may be seen as a promising possibility.

Our four-week laboratory tests (Fig. 3) suggested high removal efficiencies of the four investigated CCFS that were in

agreement with those found by Mwabi et al. (2013). However, since we used high loading, a removal rate of more than 99.9999% still resulted in up to $5 * 10^3$ CFU $l^{-1}$ in the filtrate. From a total of 45 filtrate samples only 16 (36%) were free of bacteria, as demanded by international standards for drinking water (WHO, 2011). And those were collected during the first eight days of our test. During later tests at an ambient temperature typical for field conditions, we observed considerably higher contamination of the filtrate. This means that our stepwise laboratory test actually reproduced both the impact of

higher ambient temperatures and the gradual deterioration of CCFS performance. Bacteria are removed by CCFS mainly by physical retention inside the filter candle. Thereby, the silver impregnation of the ceramic shell supports the bacterial removal by antibacterial effects (Brady et al., 2003). However, it is known that silver can be washed out and that repeated loading of a CCFS will sooner or later cause lower removal efficiencies or even a bacterial colonization inside the filter candles (Bielefeldt et al., 2009).

With this in mind, the overall failure of the CCFS during the eight months field test in Hobeni was no more a surprise. Besides the water source (springs were preferable compared to river or rain water), our field data suggested that aging was the dominant reason for poor CCFS performance. Especially in households with many children and frequent filling of the CCFS, low bacterial removal rates were observed. More frequent cleaning even showed an opposite effect and apparently increased bacterial contamination of the filtrate. This indicates that the source of contamination is not only in the top or

bottom buckets but also inside the filter candle. We did not find clear evidences for effects of poor maintenance as noticed elsewhere (e.g. Brown, 2007; Mellor et al., 2014). This suggests that in Hobeni more intense teaching of CCFS use will probably not significantly improve the situation. We are aware that in addition to the qualitative nature of the dip slide readings, the uneven distribution of explaining factors limited the explanatory strength of our data. For example, the majority



of households filled and cleaned the CCFS once a day and water temperatures mainly clustered in a range between 22 and 25 °C (Fig. 7). Nevertheless, the documented effects of water source and temperature are logic and may thus be taken as a proof for the validity of our approach.

On the whole, our data suggests that the field performance of the investigated CCFS was not satisfactory. Although the duration of the test (8 months) was well beyond the specified lifetime of the CCFS candles (12 months), not a single dip slide indicated filtrate that was free of coliforms. Owing to the limited sensitivity of the dip slide method, this suggests massive contamination ($> 10^4$ CFU l$^{-1}$) of all filtrate samples. On top of that, more than one third of the CCFS caused deterioration of hygienic water quality. These findings contradict other field tests (e.g. Brown 2007) and may thus be limited to the specific filter type we tested. However, our data suggests that not every CCFS is per se efficient and that each filter type needs a thorough check when distributed in the field.

## 6 Conclusions

This study shows that the performance of a specific, low-cost two-bucket ceramic candle filter system (CCFS) was not satisfactory for home-based water treatment in a remote rural community of Southern Africa. A stepwise laboratory test documented the combined effect of repeated loading and ambient field temperatures. In the field, the distributed CCFS already failed after an eight month use resembling only two thirds of its specified lifetime. Although users were taught how to handle and maintain the systems and the general acceptance was high, none of the distributed CCFS produced water without distinct contamination. Besides the water source (springs were preferable compared to river or rain water) our data suggests that a high water throughput was the dominant reason for a poor CCFS performance. The fact that more than one third of the investigated systems even deteriorated water quality should be regarded as an alarming sign that systems used for household water cleaning should critically be tested. Our stepwise performance test (repeated loading first at low and then at typical field temperatures) is one possibility. If systems show a significant performance loss already during such tests, a satisfactory field performance should not be expected. But also in the field the efficiency of any installed CCFS should continuously be monitored. Since the access to adequate laboratory facilities is usually restricted, dip slides may be regarded as a cost-efficient alternative to assess critical contamination even by laypersons and hence by the CCFS users themselves. Notwithstanding the obvious failure of the specific CCFS type evaluated in this study, it had a very high acceptance within the community which motivates a follow up study. Therein, alternative systems should be tested together with continuous monitoring of cleaning efficiency.



**Acknowledgements**

We express sincere thanks to all members, friends and donors of Tapini e.V. and the partner association Bulungula Incubator (BI) in South Africa, who made this study possible with a lot of personal effort in fund raising and the organizing of all logistics. Special thanks also go to all employees, residents and especially to Alexandra and Michael of Ikhaya Loxolo, who provided local accommodation. Liesl Benjamin helped a lot through cultural and linguistic mediation and massive physical effort during the time of field research in Hobeni. And finally, we thank the entire team of the hydrological lab at the University of Freiburg, especially Petra Küfner, for technical support and patience during the laboratory analysis.

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

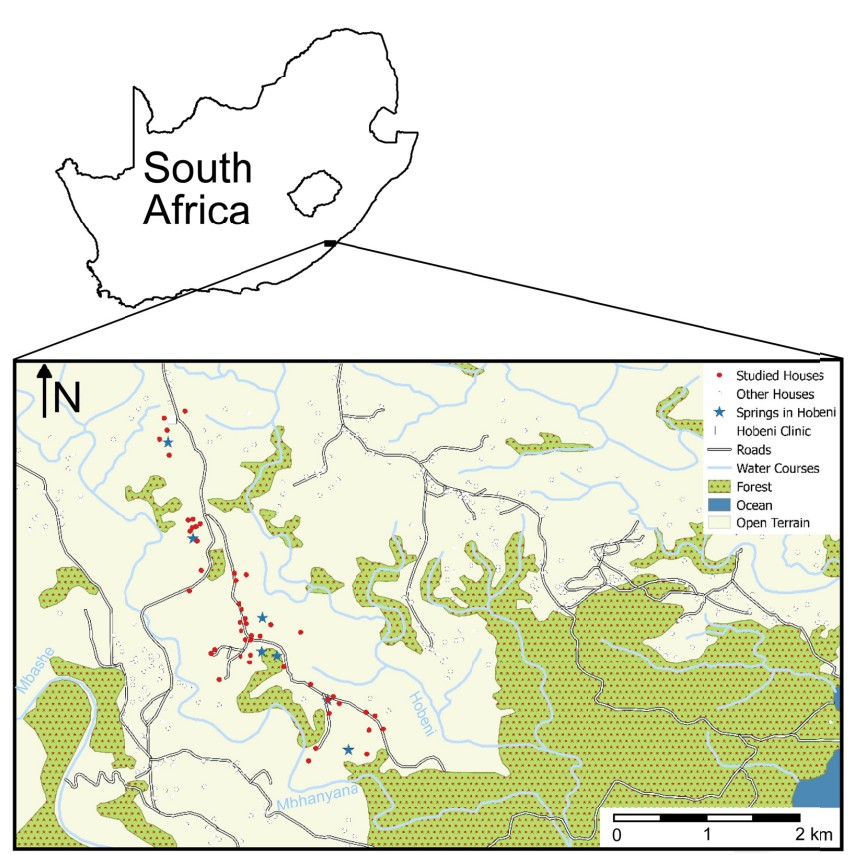

**Figure 1: Location Map.**



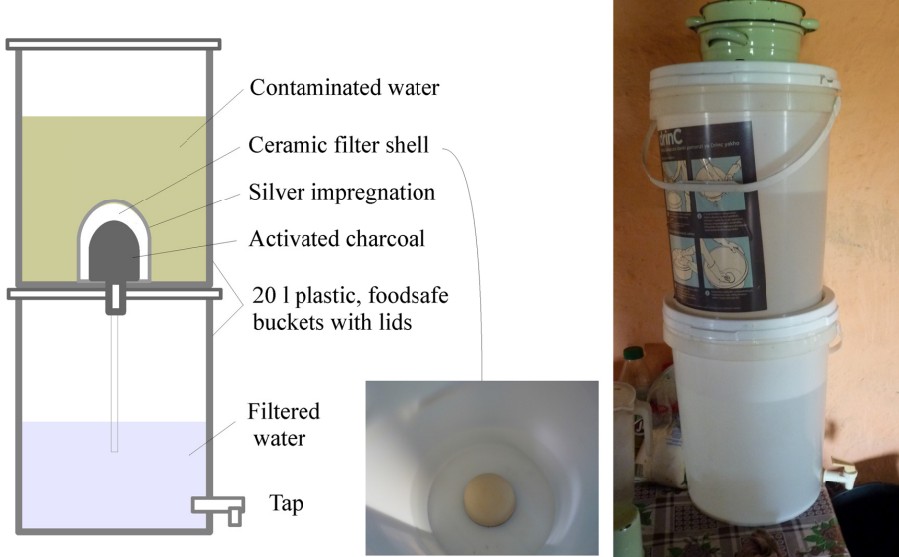

**Figure 2: Technical layout of the tested CCFS.**





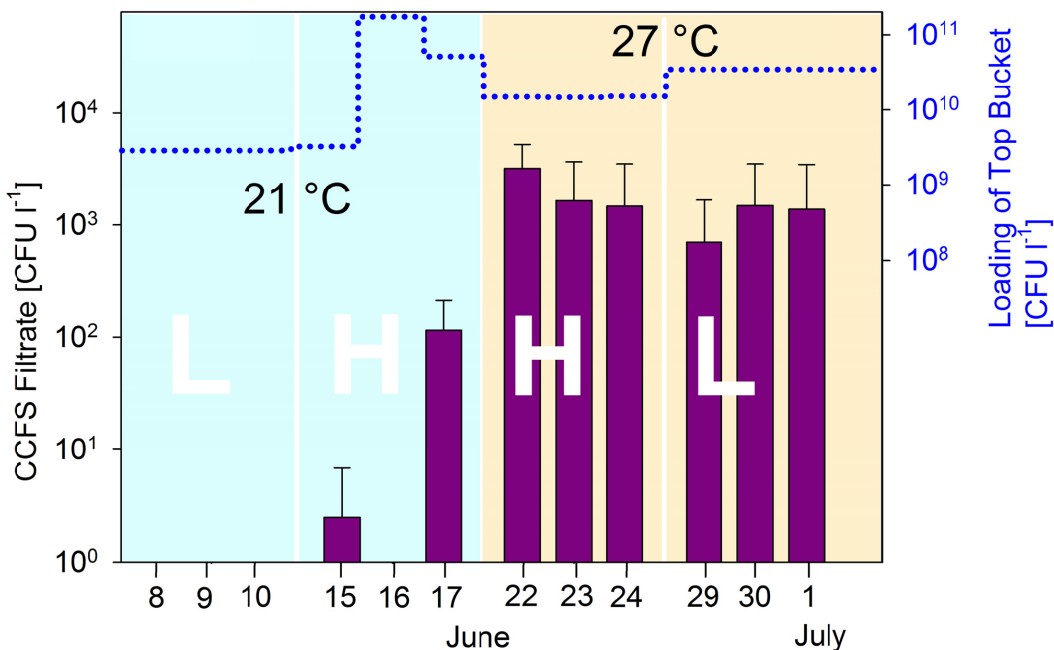

Figure 3: Stepwise laboratory performance tests; blue dotted line: loading rates of the top buckets; bars denote the mean, errors
bars the standard deviation of coliform counts in the filtrate of four replicates as detected by the membrane filter method; L
stands for low-, H for high filling scenario.





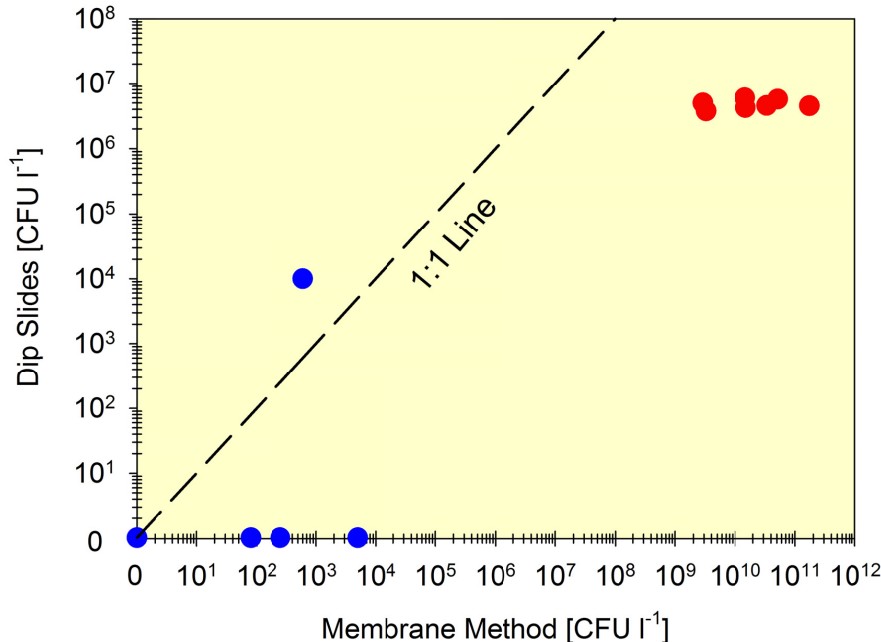

**Figure 4: Comparison of dip slides with the accredited membrane method; Blue: samples of drinking water and bottom buckets during final low filling scenario; red: samples of top buckets.**



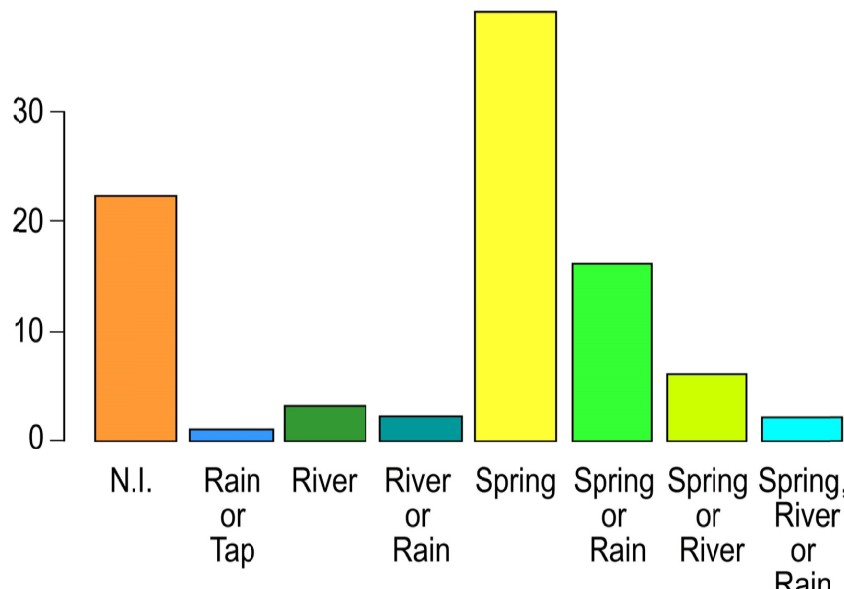

5    **Figure 5: Water sources used for the filling of the CCFS in 91 households.**




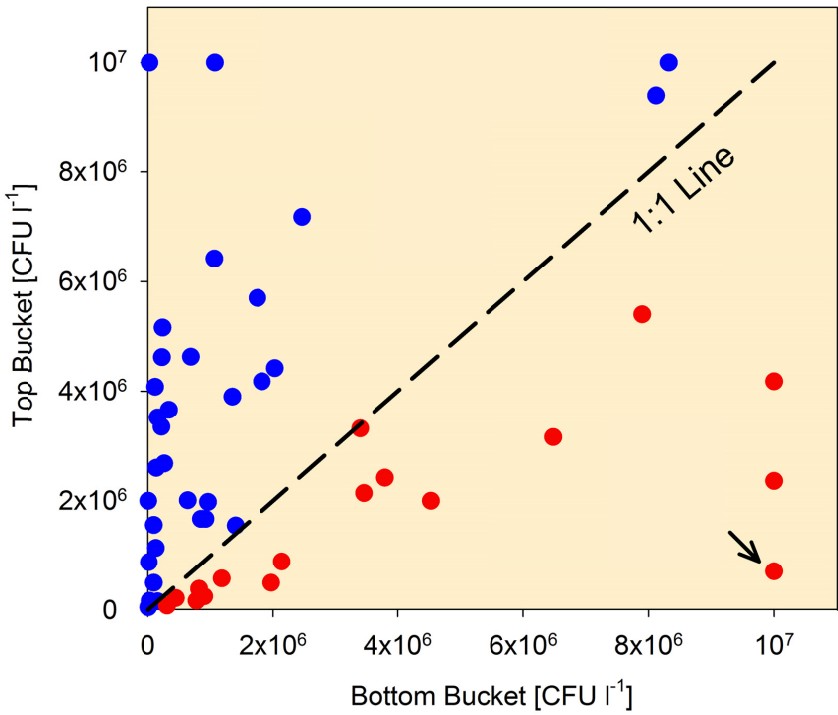

**Figure 6: Coliform counts (CFU l⁻¹) in 51 CCFS systems sampled by dip slides in Hobeni village; red dots: water quality deterioration; arrow: outlier removed for further analysis.**





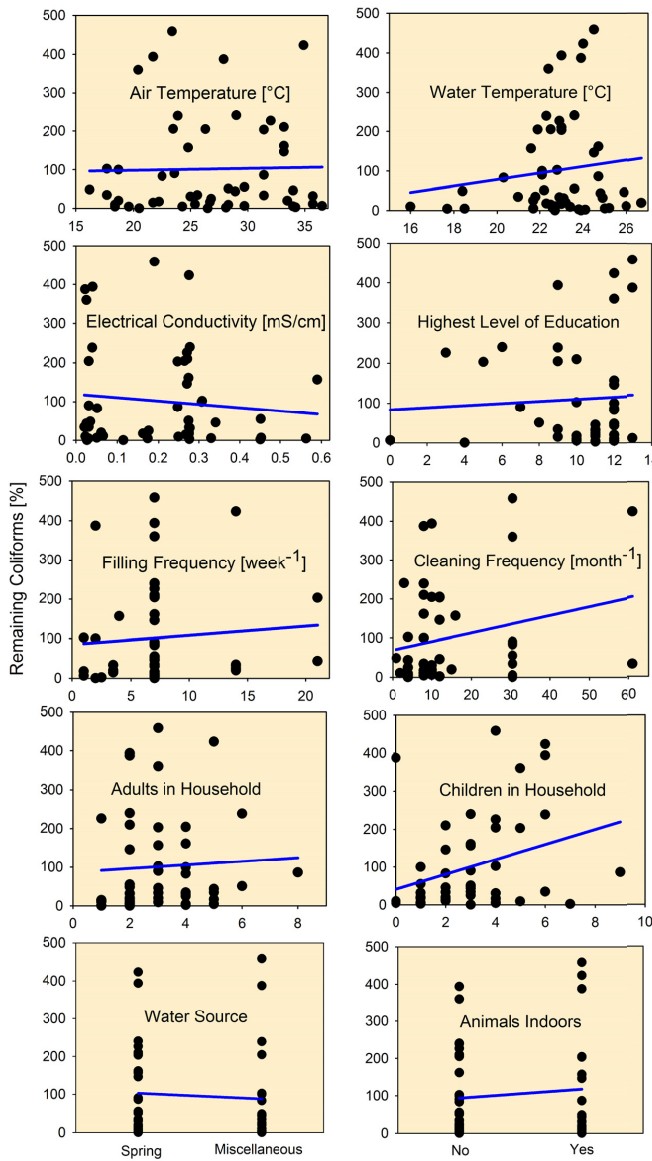

**Figure 7:** CCFS-Efficiency plotted against various factors included in the questionnaire; blue lines (linear regression) indicate the direction of influence.