# Peer review of "Do low-cost ceramic water filters improve water security in rural South Africa?"

_Drinking Water Engineering and Science, 2016_

## Referee Comment (RC1) · Anonymous Referee #1 · 27 Jul 2016

General comments: The paper reports on the performance of low-cost ceramic filters (CCFS) for improving water security in rural South Africa. This is an interesting topic considering the challenges faced by rural communities in South Africa and many other developing countries where point of use devices such as CCFS could aid in improving access to potable water in these areas. There is dire need for knowledge of the performance of such point of use devices for informed decision making while promoting the same.

Specific comments: 1. Page 2, lines 12-14-what have been the general trends in levels of acceptability of the technology, affordability, reliability in terms of amount water generated per day and willingness to pay for the CCFS in rural communities especially where these have been promoted by Du Preez et al., 2008 and Mwabi et al., 2013?

2. Page 3, line 10 is not clear. Please consider revising.

3. Section 3.1 does not sound like methodology rather a general description of the CCFS. Authors are better off explaining what was actually done with the CCFS.

4. Page 3, line 24-Did the authors also use the BiopadesLite© software to evaluate the performance of dip slides in detection of the coliforms? This is not coming out very clear here.

5. Page 4, lines 18-which type of bacteria were analysed to evaluate the performance of the NUT/AMC Dip slides in the field? What can you say about issues of bacterial regrowth in relation to issues of maintenance and performance of the CCFS?

6. Page 5, lines 2-3-which statistical test was performed to test the significance differences between the filling scenarios or immediate effects of different loading concentrations? At what significance level?

7. Page 5, line 11-what were the remaining 74% households without toilets using? Were they practicing open defecation? This might definitely have been contributed to the high levels of fecal contamination in water sources in the area. Are there any behavioral change interventions being done in the community?

8. Page 6, line 21-what platform is used for silver impregnation to improve strength and avoid silver being washed out during repeated loading of CCFS? The silver washed out could have a pollution effect as well.

9. Page 6, line 26-what is the shelf life of the CCFS? How do you explain the aging with respect to shelf life and problems arising from the maintenance of CCFS?

10. Figure 5-please label the y-axis and extend the axis so that even the value for the 'spring only' is easier to read from the axis.

---

## Referee Comment (RC2) · Anonymous Referee #2 · 27 Jul 2016

Do low-cost ceramic water filters improve water security in rural South Africa?

General comment: This study evaluates the performance of 51 ceramic candle filters (CCFS) for the production of safe drinking water in rural South Africa. The authors evaluated different factors which affect the performance of CCFS and propose the use of dip slides as cost-efficient alternative to standard laboratory tests for detection of microbial contaminants. The authors have successfully shown that the performance of the filters is affected by various factors which reduce the filters' life span. Overall the manuscript is well written. Having said that, I have the following comments for the authors: a) Page 2, line 12: Please acknowledge recent literature publications which have looked into ceramic and biosand filters. b) Page 2, line 25: What is the accuracy of dip slides? c) Page 3, section 3.1: Please add more information about the CCFS e.g. pore size, shelf life, dimensions etc. What type of contaminants do they remove?

d) Page 4, line 5: Please clarify on the filtration procedure and filtrate collection. Was the filtrate discarded after 7 h of filtration or filtration was allowed to run for 48 h with filtrate collected after 7 h, 24 h, and 48 h?  e) Page 5, line 26: The filters may be damaged during cleaning resulting in poor performance. Please explain how the filters were cleaned in the field. Is this the recommended cleaning procedure? f) Page 5, line 27: The water quality for the water sources (in terms of microbial contamination and turbidity) may differ due to seasonal variations in rainfall.  How did the water quality change for the different water sources (during rainy and dry seasons) and how did this affect the performance of CCFS? g) Page 6, line 24: Did the flow rate of the filters change over time? How does this correlate with filter performance?

Please also note the supplement to this comment:
http://www.drink-water-eng-sci-discuss.net/dwes-2016-6/dwes-2016-6-RC2-supplement.pdf

---

## Referee Comment (RC3) · Anonymous Referee #3 · 8 Aug 2016

General comments

The manuscript reports on the performance examination of a low-cost ceramic candle filter system (CCFS) for point of use (POU) drinking water treatment in the village of Hobeni, Eastern Cape Province, South Africa. The study presents an important contribution towards improving water security particularly in rural areas with disadvantaged communities like Hobeni villagers. The report emphasizes on the importance of carrying out performance monitoring programs once water treatment devices are distributed in the field rather than depending on data accumulated during laboratory tests. It is important the information presented in the manuscript to be shared among different stakeholders working in water sectors to improve means of securing water in area without centralized treatment systems. However, the authors are advised to work on the few comments below to improve the manuscript.

Specific comments

Page 1 line 8–9-The second sentence in the abstract is not well connected with the 1st and 3rd statements. It is suggested to be moved down to line 14 or deleted, and thereby the word 'moreover' in line 9 will be deleted as well.

Authors need to be specific and careful when using old information in a new statement. For example in the following cases; Page 1 line 16- replace the word 'they' with 'these slides', page 2 line 1- change the statement 'Their efficiency' to 'The efficiency of these treatment systems', page 2 line 7- replace the word 'them' with 'advanced physical methods' and the likes in the document.

Page 2 line 25- Change the word 'personal' to 'personnel'.

Page 2 line 28- Delete the word 'systems' after CCFS

Page 2 line 29- Change the word 'thereby' to 'thereafter or subsequently'.

Page 2 line 29- What were the criteria used to decide performance evaluation to be done after 8 months? Would it be possible to conduct the evaluation on monthly basis? Is there any possibility that the performance of the CCFS to be affected by seasons in a year?

Page 2 line 26—30- Authors are advised to write this paragraph in past tense. The paragraph describes what was done in their research so is better if reported in past tense with passive voice.

Page 3 line 10- Rearrange the words 'brand the name' to be read 'the brand name'

Page 3 line 15—16- The rate of 1 L/h is able to produce adequate daily drinking water volume. This is with respect to what number of family members in a household?

Page 5 line 11- If about 74 % of visited households had no access to toilet facilities, what were their practices? How have such practices affected the quality of drinking water sources? Based on this observation the authors may as well recommend for

sanitation educational campaigns and behavioral change interventions in the area.

Page 5 line 14—15- Does the absence of digestive affliction attributed to the use of CCFS? If yes, how long Hobeni people have been using CCFS? Were the authors the first to distribute CCFS or CCFS were there before this research. If test results indicated deteriorated water quality (presence of coliform bacteria) what made the communities not to have incidences of digestive afflictions? Were there any other intervention methods in the study area?

Page 5 line 16- Do the authors have any idea as to why the other 40 households abandoned the use of CCFS?

Page 6 line 9—10- The analysis procedure for dip slides as described on page 4 involves incubation of the pedals and vials after exposure time. How accessible are the incubation facilities to the CCFS household users in remote rural villages like Hobeni for them to be able to monitor the CCFS efficiency using this technology?

Page 7 line 5- Change the word 'beyond' to 'below'.

Page 14 figure 5- Authors needs to redraw the figure and extend the scale to include even the highest frequency parameters. Also y-axis needs to be labeled.

Page 16 figure 7- Authors are advised to indicate the unit for levels of education in the figure.

---

## Author Comment (AC1) · 16 Sep 2016

Answers to referee #1

General comments: The paper reports on the performance of low-cost ceramic filters (CCFS) for improving water security in rural South Africa. This is an interesting topic considering the challenges faced by rural communities in South Africa and many other developing countries where point of use devices such as CCFS could aid in improving access to potable water in these areas. There is dire need for knowledge of the performance of such point of use devices for informed decision making while promoting the same.

Our answer: We thank the anonymous referee #1 for his thorough check and his helpful comments that greatly improve our manuscript.

Specific comments:

1. Page 2, lines 12-14- what have been the general trends in levels of acceptability of the technology, affordability, reliability in terms of amount water generated per day and willingness to pay for the CCFS in rural communities especially where these have been promoted by Du Preez et al., 2008 and Mwabi et al., 2013?

Our answer: We will add the price of the bucket water filter (ZAR 599) in the filter description and further details on acceptability of this specific filter system. The study of Du Preez et al. (2008) tested CCFS performance in the field and reported their high acceptability, while the laboratory study of Mwabi et al. (2013) does not contain information about acceptability. We will add the following sentence:

"Du Preez et al. (2008) reported about high acceptability of CCFS in rural communities of South Africa."

We did not include questions on willingness to pay in our questionnaire, but we think that a recent study by Rananga and Gumbo (2015) in another South African municipality may provide useful information. So we will additionally add the following sentence:

"The need for improved drinking water supply in South Africa was recently manifested for two communities in the Municipality of Mutale (Rananga and Gumbo, 2015): 95% of the households were willing to pay for reliable drinking water supply, those with tertiary level education would afford ZAR 150 per month."

Reference: Rananga, H.T., Gumbo, J.R.: Willingness to Pay for Water Services in Two Communities of Mutale Local Municipality, South Africa: A Case Study, Journal of Human Ecology, 49(3), 231-243, 2015.

2. Page 3, line 10 is not clear. Please consider revising.

Our answer: We will revise the sentence as follows:

"We selected a widely used, low-cost (ZAR 599,-) two-bucket CCFS for our laboratory

[Figure]

and field tests. It is commercially distributed under the brand name DrinC by Headstream Pure Water, Johannesburg, South Africa."

3. Section 3.1 does not sound like methodology rather a general description of the CCFS. Authors are better off explaining what was actually done with the CCFS.

Our answer: With the two new introductory sentences (see 2. above), we hope that it is now clear what was actually done with the CCFS in section 3.1. Also recommended by reviewer #2, we will also add more information about the filters as follows:

"The ceramic filter candles consist of a 0.2 $\mu$m silver-impregnated ceramic shell containing an activated charcoal interior medium. The filters have a diameter of 0.1 m and unlimited shelf life. Once in use, the filter candles have to be replaced once a year (DrinC, 2016). Raw water is filled into the top bucket. The water drips through the candle filter unit into the bottom bucket, where clean water can be drained through the tap. According to the manufacturer CCFS remove >99.9% of harmful bacteria (100% of E. Coli), >98% of particles larger than 0.2 $\mu$m, >96% of metals like Fe, Al, Pb, and >80% of various organic pollutants."

4. Page 3, line 24-Did the authors also use the BiopadesLite[©] software to evaluate the performance of dip slides in detection of the coliforms? This is not coming out very clear here.

Our answer: We did not use this software. We will add the following sentence:

"The BiopadesLite[©] software had not been available to be used in the present study."

5. Page 4, lines 18-which type of bacteria were analysed to evaluate the performance of the NUT/AMC Dip slides in the field?

Our answer: We analysed aerobic coliform bacteria. So we added the following sentence:

"Our dip slides detected aerobic coliform bacteria as dots of a red-colored dye."

What can you say about issues of bacterial regrowth in relation to issues of maintenance and performance of the CCFS?

Our answer: Also recommended by reviewer #2, we will add the following sentences about the recommended cleaning procedure in the method section:

"Users are advised to clean the filter every time the water flow becomes too slow. Then the bottom bucket should be cleaned by a bleach solution and the filter candle by a non-metal scrubbing pad."

In addition we updated figure 7 to also include the cleaning method. There was no influence on using the recommended bleach solution, the filters even deteriorated when bleaching was applied. We will add the following sentence:

"Also recommended bleaching did not improve CCFS performance."

6. Page 5, lines 2-3-which statistical test was performed to test the significance differences between the filling scenarios or immediate effects of different loading concentrations? At what significance level?

Our answer: Due to your comment we noticed that the formulation regarding the significance of the filling scenarios is unclear. Strictly speaking, the filling scenarios could not be tested, because they were influenced by the repeated loading. We also provide information about the used test procedure. We will re-formulate as follows:

"We fitted regressions to our data using the Generalized Linear Model with significance level $p < 0.05$ for hypothesis testing. Those revealed no significant difference of different loading concentrations. The filling scenarios were influenced by repeated loading that had a significant influence on CCFS performance."

7. Page 5, line 11-what were the remaining 74% households without toilets using? Were they practicing open defecation? This might definitely have been contributed to the high levels of fecal contamination in water sources in the area. Are there any behavioral change interventions being done in the community?

Our answer: The rest of the households were indeed practicing open defecation at that time, which definitely can increase the fecal contamination of the water sources. We are not aware of behavioural change interventions in the area but will recommend those, also following referee #3. Open defecation was one reason, why we started our project and distributed CCFS in the area. We will add the following sentences:

"The remaining 74% of the households were practicing open defecation, which must be considered as a serious threat for hygienic drinking water quality. This was one of the reasons why CCFS were distributed in Hobeni."

8. Page 6, line 21-what platform is used for silver impregnation to improve strength and avoid silver being washed out during repeated loading of CCFS? The silver washed out could have a pollution effect as well.

Our answer: Also recommended by reviewer #2, we will add more details of the filter candle, also on the platform where the silver impregnation is attached. In addition we will add a reference on silver-ceramic composites and a short discussion on silver wash-out and environmental and human toxicity of silver nanoparticles:

"The ceramic filter candles consist of a 0.2 $\mu$m silver-impregnated ceramic shell containing an activated charcoal interior medium. Lv et al. (2009) showed that silver nanoparticle–porous ceramic composites show efficient antibacterial effects without a measurable loss of nanoparticles. However, incorporated into water filters, Bielefeld et al (2009) documented a significant wash-out of silver with decreasing filter efficiency. This process has to be taken into account for this type of water filter and per se causes a limited life time. Silver nanoparticles are widely used in various biomedical applications, although it is difficult to draw definite conclusions about their human and environmental toxicity (Wei et al. 2015)."

References: Bielefeldt, A.R., Kowalski, K., and Summers, R.S.: Bacterial treatment effectiveness of point-of-use ceramic water filters, Water Res. 43, 3559-3565, doi:10.1016/j.watres.2009.04.047, 2009. Lv, Y., Liu, H., Wang, Z., Liu, S., Hao, L.,

Sang, Y., Liu, D., Wang, J., Boughton, R.I.: Silver nanoparticle-decorated porous ceramic composite for water treatment, Journal of Membrane Science 331, 50–56, 2009.
Wei, L, Lu, J., Xu, H., Patel, A., Chen, Z-S., Chen, G.: Silver nanoparticles: synthesis, properties, and therapeutic applications, Drug Discovery Today 20 (5), 595-601, 2015.

9. Page 6, line 26-what is the shelf life of the CCFS?

Our answer: According to the manufacturer the shelf life is unlimited and during use, the candles have to be replaced once a year. This will be written into section 3.1:

"The filter candles have a diameter of 0.1 m and unlimited shelf life (JustWater 2016). Once in use, the candles have to be replaced once a year (DrinC, 2016)."

References DrinC: Instructions DrinC Water Bucket, http://drinc.co.za/instructions/water-bucket, last access: 16 September 2016. Just-Water: 4"X4" Ceramic Filter, http://www.justwater.me/products/4x4-ceramic-filter, last access: 16 September 2016.

How do you explain the aging with respect to shelf life and problems arising from the maintenance of CCFS?

Our answer: Also requested by referee #2 we will add the following sentences about the recommended cleaning/maintenance procedure in the method section:

"Users are advised to clean the filter every time the water flow becomes too slow. Then the bottom bucket should be cleaned by a bleach solution and the filter candle by a non-metal scrubbing pad."

In addition we will update figure 7 to also include the cleaning method. There it will be shown that there was no influence using the recommended bleach solution. We will add the following sentence:

"Also recommended bleaching did not improve CCFS performance."

10. Figure 5-please label the y-axis and extend the axis so that even the value for the

'spring only' is easier to read from the axis.

Our answer: Figure 5 will be modified with extended axis. It is attached
* * *
[Figure]

**Fig. 1.** Updated figure 5

---

## Author Comment (AC2) · 16 Sep 2016

Answers to referee #2

General comment: This study evaluates the performance of 51 ceramic candle filters (CCFS) for the production of safe drinking water in rural South Africa. The authors evaluated different factors which affect the performance of CCFS and propose the use of dip slides as cost-efficient alternative to standard laboratory tests for detection of microbial contaminants. The authors have successfully shown that the performance of the filters is affected by various factors which reduce the filters' life span. Overall the manuscript is well written. Having said that, I have the following comments for the authors:

Our answer: We also thank the anonymous referee #2 for his thorough check. Also his

comments helped us to improve our manuscript.

Specific comments

a) Page 2, line 12: Please acknowledge recent literature publications which have looked into ceramic and biosand filters.

Our answer: We will add the following paragraph including recent literature on ceramic and biosand filters:

"Murphy et al. (2010) monitored the microbiological and chemical quality of treated water from BSF and CWF in rural Cambodia. During a six month period both technologies improved water quality for various parameters but failed to consistently meet the WHO drinking water guidelines for nitrite and for E. coli. BSF can be improved by using iron oxide coated sand (Ahammed and Davra, 2011) or by adding a layer of bark biomass (Ali Baig et al., 2011). Also continuous operation resulted in better BSF performance (Young-Rojanschi and Madramootoo, 2014). Other studies concentrated on cost-efficient POU water treatment techniques. BSF with plastic housing were found to be as efficient as their concrete counterparts (Fabiszewski de Aceituno et al., 2012) and various researchers reported the success of locally produced, low-cost CWF (Brown et al. 2008, Simonis and Basson, 2011, Mwabi et al. 2013). Recent reviews about reduction of diarrhoea by different POU interventions found filtration techniques superior to solar or chemical water disinfection (Wolf et al., 2014, Clasen et al., 2015). However, comparisons were blurred by a high risk of bias, since the data relied on self-reported diarrhea and placebo POU interventions were missing."

References: Ahammed, M.N., Davra, K.: Performance evaluation of biosand filter modified with iron oxide-coated sand for household treatment of drinking water, Desalination 276, 287–293, 2011, doi:10.1016/j.desal.2011.03.065.

Ali Baig, S., Mahmood,Q., Nawab, B., Nawaz, B., Shafqat, M.N., Pervez, A. (2011): Improvement of drinking water quality by using plant biomass through household

biosand filter – A decentralized approach, Ecological Engineering 37, 1842– 1848, 2011, doi:10.1016/j.ecoleng.2011.06.011.

Brown, J., Sobsey, M.D., Loomis, D.: Local drinking water filters reduce diarrheal disease in Cambodia: a randomized, controlled trial of the ceramic water purifier, American Journal of Tropical Medicine and Hygiene, 79(3), 394–400, 2008.

Clasen, T.F., Alexander, K.T., Sinclair, D., Boisson, S., Peletz, R., Chang, H.H, Majorin, F., Cairncross, S.: Interventions to improve water quality for preventing diarrhoea, Cochrane Database of Systematic Reviews 2015, Issue 10. Art. No.: CD004794, 2015, DOI: 10.1002/14651858.CD004794.pub3.

Fabiszewski de Aceituno, A.M., Stauber, C.E., Walters, A.R., Meza Sanchez, R.E., Sobsey, M.D.: A randomized controlled trial of the plastic-housing BioSand filter and its impact on diarrheal disease in Copan, Honduras. American Journal of Tropical Medicine and Hygiene, 86 (6), 913–21, 2012.

Murphy, H.M., McBean, E.A., Farahbakhsh, K.: A critical evaluation of two point-of-use water treatment technologies: can they provide water that meets WHO drinking water guidelines?, Journal of Water and Health 8 (4), 611-630, 2010, doi: 10.2166/wh.2010.156.

Mwabi, J. K., Mamba, B. B., and Momba M. N. B.: Removal of waterborne bacteria from surface water and groundwater by cost-effective household water treatment systems (HWTS): A sustainable solution for improving water quality in rural communities of Africa, Water SA, 39 (4), 445-456, 2013.

Shams Ali Baig, S., Mahmood, Q., Nawab, B., Shafqat, M.N., Pervez, A.: Improvement of drinking water quality by using plant biomass through household biosand filter – A decentralized approach, Ecological Engineering 37, 1842– 1848, 2011.

Simonis, J.J., Basson, A.K.: Evaluation of a low-cost ceramic micro-porous filter for elimination of common disease microorganisms, Physics and Chemistry of the Earth,

36, 1129–1134, 2011, doi:10.1016/j.pce.2011.07.064.

Wolf, J., Prüss-Ustün, A., Cumming, O., Bartram, J., Bonjour, S., Cairncross, S., Clasen, T., Colford, J. M., Curtis, V., De France, J., Fewtrell, L., Freeman, M.C., Gordon, B., Hunter, P.R., Jeandron, A., Johnston, R.B., Mäusezahl, D., Mathers, C., Neira, M., Higgins, J.P.T.: Assessing the impact of drinking water and sanitation on diarrhoeal disease in low- and middle-income settings: systematic review and meta-regression, Tropical Medicine and International Health 19 (8), 928–942, 2014, doi:10.1111/tmi.12331.

Young-Rojanschi, C., Madramootoo, C.: Intermittent versus continuous operation of biosand filters, Water Research 49, 1-10, 2014, doi.:10.1016/j.watres.2013.11.011.

b) Page 2, line 25: What is the accuracy of dip slides?

Our answer: We will include two references on the accuracy of dip slides (for contact methods and for water):

"Dip slides had similar precision as swapping or contact agar plates during detection of contamination on artificially soiled stainless-steel surfaces (Salo et al., 2000). For drinking water, dip slides had considerably less accuracy than membrane filter methods but were recommended for the detection of massive contamination of drinking water sources (Vanderzwaag et al., 2009)."

References:

Salo, S., Laine, A., Alanko, T., Sjöberg, A.-M., Wirtanen, G.: Journal of AOAC International, 83(6), 1357-1366, 2000.

Vanderzwaag, J.C., Bartlett, K.H., Atwater, J.W., Baker, D.: Evaluation of Field Testing Techniques Used in a Household Water Treatment Study in Posoltega, Nicaragua, Water Quality Research Journal of Canada, 44, 122-131, 2009.

c) Page 3, section 3.1: Please add more information about the CCFS e.g. pore size, shelf life, dimensions etc. What type of contaminants do they remove?

Our answer: We will add more information about the filters as follows:

"The filter candles have a diameter of 0.1 m and unlimited shelf life (JustWater 2016). Once in use, the candles have to be replaced once a year (DrinC, 2016). Raw water is filled into the top bucket. The water drips through the candle filter unit into the bottom bucket, where clean water can be drained through the tap. According to the manufacturer CCFS remove >99.9% of harmful bacteria (100% of E. Coli), >98% of particles larger than 0.2 $\mu$m, >96% of metals like Fe, Al, Pb, and >80% of various organic pollutants."

References:

DrinC: Instructions DrinC Water Bucket, http://drinc.co.za/instructions/water-bucket, last access: 16 September 2016.

JustWater: 4"X4" Ceramic Filter, http://www.justwater.me/products/4x4-ceramic-filter, last access: 16 September 2016.

d) Page 4, line 5: Please clarify on the filtration procedure and filtrate collection. Was the filtrate discarded after 7 h of filtration or filtration was allowed to run for 48 h with filtrate collected after 7 h, 24 h, and 48 h?

Our answer: We will modify the sentence to clarify the filtration procedure:

"The top bucket of the CCFS was filled once, allowed to run for 48 h with 100 ml of filtrate collected from the bottom bucket after 7 h, 24 h, and 48 h. During this 48 h the top bucket was not filled up again."

e) Page 5, line 26: The filters may be damaged during cleaning resulting in poor performance. Please explain how the filters were cleaned in the field. Is this the recommended cleaning procedure?

Our answer: Also requested by referee #1, we will add the following sentences about the recommended cleaning procedure in the method section:

"Users are advised to clean the filter every time the water flow becomes too slow. Then the bottom bucket should be cleaned by a bleach solution and the filter candle by a non-metal scrubbing pad."

In addition we will update figure 7 (attached) to also include the cleaning method. There was no influence on using the recommended bleach solution, the filters even deteriorated when bleaching was applied. We will add the following sentence:

"Also recommended bleaching did not improve CCFS performance."

f) Page 5, line 27: The water quality for the water sources (in terms of microbial contamination and turbidity) may differ due to seasonal variations in rainfall. How did the water quality change for the different water sources (during rainy and dry seasons) and how did this affect the performance of CCFS?

Our answer: We will include a graph on average climate into figure 1. We cannot exclude seasonal differences in water quality of the different water sources. Moreover, we assume that people in the area collect their drinking water from different sources in different seasons, which may also affect CCFS performance. To study these effects, measurements on a regular basis would be necessary that we recommend in the discussion: We will add the following sentence:

"Monitoring of CCFS performance should be carried out on a monthly basis to also include seasonal changes in water quality"

g) Page 6, line 24: Did the flow rate of the filters change over time? How does this correlate with filter performance?

Our answer: We do not have continuous data about flow rates, but at the end of our test, only 1 % of the households complained about slow filtering. So in most households the original flow rates could be re-established by cleaning as recommended. We will add the recommended cleaning procedure that should avoid blocking of the filters (see e) above) and will add a sentence on the results of the field survey showing the fact that

most users were satisfied with the flow rate:

"Only 4.4 % of the households complained about the intensive maintenance of the CCFS, 3.3 % about the long distances to the water sources and only 1.1% about a slow filtering time.

[Figure]

**Fig. 1.** Modified Figure 1

[Figure]

**Fig. 2.** Updated Figure 7

---

## Author Comment (AC3) · 16 Sep 2016

Answers to referee #3

General comments The manuscript reports on the performance examination of a low-cost ceramic candle filter system (CCFS) for point of use (POU) drinking water treatment in the village of Hobeni, Eastern Cape Province, South Africa. The study presents an important contribution towards improving water security particularly in rural areas with disadvantaged communities like Hobeni villagers. The report emphasizes on the importance of carrying out performance monitoring programs once water treatment devices are distributed in the field rather than depending on data accumulated during laboratory tests. It is important the information presented in the manuscript to be shared among different stakeholders working in water sectors to improve means of securing water in area without centralized treatment systems. However, the authors are advised to work on the few comments below to improve the manuscript.

Our answer: We also thank the anonymous referee #3 for his thorough check and his constructive comments. This greatly helped to improve our manuscript.

Specific comments

Page 1 line 8–9-The second sentence in the abstract is not well connected with the 1st and 3rd statements. It is suggested to be moved down to line 14 or deleted, and thereby the word 'moreover' in line 9 will be deleted as well.

Our answer: We will change the abstract accordingly.

Page 2 line 25- Change the word 'personal' to 'personnel'.

Our answer: We will change this accordingly.

Page 2 line 28- Delete the word 'systems' after CCFS

Our answer: We will change this accordingly.

Page 2 line 29- Change the word 'thereby' to 'thereafter or subsequently'.

Our answer: We will change this accordingly.

Page 2 line 29- What were the criteria used to decide performance evaluation to be done after 8 months? Would it be possible to conduct the evaluation on monthly basis? Is there any possibility that the performance of the CCFS to be affected by seasons in a year?

Our answer: We decided to evaluate performance after 8 months, because this equals two thirds of the specified lifetime of the filter candles. Hence the systems should still work efficiently while malfunctions should turn out clearly. It is true that more frequent testing is advisable also to show the impacts of seasons, which was also proposed by referee #2. This we will recommend in the discussion section:

"Monitoring of CCFS performance should be carried out on a monthly basis to also include seasonal changes in water quality"

Page 2 line 26-30- Authors are advised to write this paragraph in past tense. The paragraph describes what was done in their research so is better if reported in past tense with passive voice.

Our answer: We will change this accordingly.

Page 3 line 10- Rearrange the words 'brand the name' to be read 'the brand name'

Our answer: We will change this accordingly.

Page 3 line 15-16- The rate of 1 L/h is able to produce adequate daily drinking water volume. This is with respect to what number of family members in a household?

Our answer: In general, CCFS have flow rates of approximately 1 L/h and produce 10 L of drinking water per day according to CAWST(2011). The specific CCFS tested by us can reach hourly rates of up to 4 L/h (Mwabi et al, 2013). This results in approximately 40 L/day that should suffice for an average household size of 6 persons in Hobeni. We will also provide a reference on human water need. We will re-formulate as follows:

In general, CCFS systems have flow rates of approximately 1 l h-1, depending on the batch volume (CAWST, 2011). The specific CCFS tested in this study can reach rates of up to 4 l h-1 (Mwabi et al., 2013) making up approximately 40 l d-1. This volume can be regarded adequate for an average household of 6 family members in Hobeni, if a 3-4 l need of clean drinking water per person and day is assumed (Sawka et al., 2005).

Reference: Sawka, M.N., Cheuvront, S.N., Carter, R.: Human Water Needs, Nutrition Reviews 63, 30-39, 2005.

Page 5 line 11- If about 74 % of visited households had no access to toilet facilities, what were their practices? How have such practices affected the quality of drinking water sources? Based on this observation the authors may as well recommend for

sanitation educational campaigns and behavioral change interventions in the area.

Our answer: A valuable comment, see also our answers to referee #1 where we wrote:

The rest of the households were indeed practicing open defecation at that time, which definitely can increase the fecal contamination of the water sources. We are not aware of behavioural change interventions in the area but will recommend those. Open defecation was one reason, why we started our project and distributed CCFS in the area. We will add the following sentences:

"The remaining 74% of the households were practicing open defecation, which must be considered as a serious threat for hygienic drinking water quality. This was one of the reasons why CCFS were distributed in Hobeni."

We will also add this sentence in the conclusions: "We propose sanitation educational campaigns and behavioral change interventions to complement POU water treatment in Hobeni."

Page 5 line 14-15- Does the absence of digestive affliction attributed to the use of CCFS? If yes, how long Hobeni people have been using CCFS? Were the authors the first to distribute CCFS or CCFS were there before this research. If test results indicated deteriorated water quality (presence of coliform bacteria) what made the communities not to have incidences of digestive afflictions? Were there any other intervention methods in the study area?

Our answer: We are not aware of other intervention campaigns to improve water security in Hobeni at that time and we were the first to distribute CCFS systems in this quantity. The low number of digestive afflictions (only 16% reported afflictions during the past five years in the survey) might also be due to the fact that those depend on self-reporting. As already stated in the newly added paragraph of relevant literature (requested by referee #2), self-reporting may produce substantial bias in performance tests of POU intervention methods. We will add the following sentence:

".…although morbidity rates depend on self-reporting from the household members that is known to produce substantial bias (Wolf et al., 2014, Clasen et al., 2015)."

References:

Clasen, T.F., Alexander, K.T., Sinclair, D., Boisson, S., Peletz, R., Chang, H.H, Majorin, F., Cairncross, S.: Interventions to improve water quality for preventing diarrhoea, Cochrane Database of Systematic Reviews 2015, Issue 10. Art. No.: CD004794, 2015, DOI: 10.1002/14651858.CD004794.pub3.

Wolf, J., Prüss-Ustün, A., Cumming, O., Bartram, J., Bonjour, S., Cairncross, S., Clasen, T., Colford, J. M., Curtis, V., De France, J., Fewtrell, L., Freeman, M.C., Gordon, B., Hunter, P.R., Jeandron, A., Johnston, R.B., Mäusezahl, D., Mathers, C., Neira, M., Higgins, J.P.T.: Assessing the impact of drinking water and sanitation on diarrhoeal disease in low- and middle-income settings: systematic review and meta-regression, Tropical Medicine and International Health 19 (8), 928–942, 2014, doi:10.1111/tmi.12331.

Page 5 line 16- Do the authors have any idea as to why the other 40 households abandoned the use of CCFS?

Our answer: We will include information from our survey and rewrite both the relevant paragraphs in the methodology and the result sections as follows:

Methodology (3.4): "At 51 of the visited households water from the CCFS could be tested. The remaining 40 units did not contain enough water for the testing procedure and only the survey was conducted."

Results (4.2): ".…Approximately eight months after distribution, 69 % (63 units) of the CCFS were still in regular use, 20% (18 units) were broken and 5.5% (5 units) of the households refused to use the filters due to different reasons. In 5.5% of the households (5 units) the CCFS were used only temporarily. The majority of the households (60 %) liked the clean water after the filtering procedure....."

Page 6 line 9-10- The analysis procedure for dip slides as described on page 4 involves

incubation of the pedals and vials after exposure time. How accessible are the incubation facilities to the CCFS household users in remote rural villages like Hobeni for them to be able to monitor the CCFS efficiency using this technology?

Our answer: We used a cheap, portable incubator for animal eggs to incubate the dip slides directly in the field. This can principally be used by rural communities themselves. This we will state in the manuscript:

"For the incubation of dip slides we used a cost-effective, portable, ventilated animal egg incubator with low energy consumption (220V-240V, <60 W, ZJchao)."

Page 7 line 5- Change the word 'beyond' to 'below'.

Our answer: We will change this accordingly.

Page 14 figure 5- Authors needs to redraw the figure and extend the scale to include even the highest frequency parameters. Also y-axis needs to be labeled.

Our answer: We will change this accordingly.

Page 16 figure 7- Authors are advised to indicate the unit for levels of education in the figure.

Our answer: The units are years spent for education. We will include this in the updated figure 7.
* * *
[Figure]

**Fig. 1.** Updated figure 5

[Figure]

**Fig. 2.** Updated Figure 7